# Health service utilization in immigrants with multiple sclerosis

**Dalia L. Rotstein**[1,2]*, **Ruth Ann Marrie**[3], **Karen Tu**[4,5,6], **Susan E. Schultz**[7], **Kinwah Fung**[7], **Colleen J. Maxwell**[7,8]

**1** Department of Medicine, University of Toronto, Toronto, Ontario, Canada, **2** St. Michael's Hospital, Toronto, Ontario, Canada, **3** Departments of Medicine and Community Health Sciences, Max Rady College of Medicine, Rady Faculty of Health Sciences, University of Manitoba, Winnipeg, Canada, **4** North York General Hospital, Toronto, Ontario, Canada, **5** Departments of Family and Community Medicine and Institute for Health Policy and Management, University of Toronto, Toronto, Ontario, Canada, **6** Toronto Western Hospital University Health Network, Toronto, Ontario, Canada, **7** ICES, Toronto, Ontario, Canada, **8** Schools of Pharmacy and Public Health and Health Systems, University of Waterloo, Waterloo, ON, Canada

* dalia.rotstein@unityhealth.to

## Abstract

### Background

Access to neurology specialty care can influence outcomes in individuals with multiple sclerosis (MS), but may vary based on patient sociodemographic characteristics, including immigration status.

### Objective

To compare health services utilization in the year of MS diagnosis, one year before diagnosis and two years after diagnosis in immigrants versus long-term residents in Ontario, Canada.

### Methods

We identified incident cases of MS among adults aged 20–65 years by applying a validated algorithm to health administrative data in Ontario, Canada, a region with universal health insurance and comprehensive coverage. We separately assessed hospitalizations, emergency department (ED) visits, outpatient neurology visits, other outpatient specialty visits, and primary care visits. We compared rates of health service use in immigrants versus long-term residents using negative binomial regression models with generalized estimating equations adjusted for age, sex, socioeconomic status, urban/rural residence, MS diagnosis calendar year, and comorbidity burden.

### Results

From 2003 to 2014, there were 13,028 incident MS cases in Ontario, of whom 1,070 (8.2%) were immigrants. As compared to long-term residents, rates of hospitalization were similar (Adjusted rate ratio (ARR) 0.86; 95% CI: 0.73–1.01) in immigrants the year before MS diagnosis, but outpatient neurology visits (ARR 0.93; 95% CI: 0.87–0.99) were slightly less frequent.

**Data Availability Statement:** The dataset used in this study is held securely in coded format at ICES. ICES is a prescribed entity under section 45 of Ontario's Personal Health Information Protection Act. Section 45 authorizes ICES to collect personal

health information, without consent, for the purpose of analysis of compiling statistical information with respect to the management of, evaluation or monitoring of, the allocation of resources to or planning for all or part of the health system. Legal restrictions and data sharing agreements prohibit ICES from making the dataset publicly available. Access may be granted to those who meet the conditions for confidential access, available at https://www.ices.on.ca/DAS. CM holds an appointment as an ICES Scientist, which enabled access to ICES data. Data access is available to external public sector researchers either through collaboration with an ICES scientist or directly, following project approval, via a secure online desktop infrastructure (see above link for details).

**Funding:** This study was funded through grants from the Multiple Sclerosis Society of Canada and the Consortium of Multiple Sclerosis Centers. The funding sources had no role in study design, data collection, data analysis, data interpretation, or writing of the report. All authors had full access to the data in the study and the corresponding author had the final responsibility for the decision to submit for publication. The sponsors played no role in the study design; in the collection, analysis and interpretation of data; in the writing of the report; or in the decision to submit the report for publication. This study was supported by ICES, which is funded by an annual grant from the Ontario Ministry of Health and Long-Term Care (MOHLTC). The opinions, results and conclusions reported in this paper are those of the authors and are independent from the funding sources. No endorsement by ICES or the Ontario MOHLTC is intended or should be inferred. Parts of this material are based on data and information compiled and provided by the Canadian Institute for Health Information (CIHI). However, the analyses, conclusions, opinions and statements expressed herein are those of the authors, and not necessarily those of CIHI. The funders provided support in the form of salaries for authors [SES, KF], but did not have any additional role in the study design, data collection and analysis, decision to publish, or preparation of the manuscript.

**Competing interests:** Dalia Rotstein has received research support from the Consortium of Multiple Sclerosis Centers (CMSC), the Multiple Sclerosis Society of Canada, and Roche Canada. She has served as a speaker or consultant for Alexion, Biogen, EMD Serono, Novartis, Roche and Sanofi Aventis. Ruth Ann Marrie receives research funding from: CIHR, Research Manitoba, Multiple Sclerosis Society of Canada, Multiple Sclerosis Scientific

However, immigrants had higher rates of hospitalization during the diagnosis year (ARR 1.20, 95% CI: 1.04–1.39), and had greater use of outpatient neurology (ARR 1.17, 95% CI: 1.12–1.23) but fewer ED visits (ARR 0.86; 95% CI: 0.78–0.96). In the first post-diagnosis year, immigrants continued to have greater numbers of outpatient neurology visits (ARR 1.16; 95% CI: 1.10–1.23), but had fewer hospitalizations (ARR 0.79; 95% CI: 0.67–0.94).

## Conclusions

Overall, our findings were reassuring concerning health services access for immigrants with MS in Ontario, a publicly funded health care system. However, immigrants were more likely to be hospitalized despite greater use of outpatient neurology care in the year of MS diagnosis. Reasons for this may include more severe disease presentation or lack of social support among immigrants and warrant further investigation.

## Introduction

Health service access and utilization are known to vary substantially with sociodemographic factors such as sex, race, immigrant status, and income level [1–4]. Disparities in health service utilization are reflected in divergent experiences of the health care system and outcomes [1,5]. However, health service utilization patterns have been less extensively described in the context of neurologic diseases, although one American study found that blacks were more likely to be hospitalized, and less likely to experience an ambulatory neurology visit, than whites [6]. Health service utilization patterns may be particularly relevant for multiple sclerosis (MS), a chronic inflammatory disease of the central nervous system (CNS) and one of the primary causes of disability in young adults [7]. The complex treatment landscape for MS today requires specialized neurology care and early implementation of disease-modifying therapy (DMT) to promote better long-term outcomes [7,8]. Little is known about which sociodemographic factors predict favourable health service utilization in people living with multiple sclerosis (MS). In a previous scoping review of the literature, non-white race was associated with worse access to health services for people with MS, although only 3 of 36 included studies were cohort studies [9].

A favourable pattern of health system utilization in those with established MS diagnoses would be characterized by regular outpatient neurology visits, as comprehensive MS care requires periodic monitoring of the patient's symptoms, neurologic examination, MRI, and response to DMT. Most MS attacks can be managed on an outpatient basis when there is consistent access to a neurologist; hence fewer emergency department visits and hospitalizations would be expected under favourable conditions [10]. However, the initial symptoms or attack in MS may involve an emergency department visit to facilitate diagnostic work-up and possible treatment with corticosteroids [11]. Hospitalization is usually required only for more severe attacks, the risk of which is reduced by DMT [10]. Regular primary care visits may be appropriate if they complement rather than substitute for specialty neurology care and should be interpreted in that context [12].

There has been no previous study of the effect of immigrant status on health service utilization among people with MS, which is the focus of this study. The effect of immigrant status on health service utilization is distinct from the question of race, as health system literacy, language, and cultural factors affect immigrants' interaction with the health care system [13]. We

Foundation, Crohn's and Colitis Canada, National Multiple Sclerosis Society, CMSC and United States Department of Defence. She is supported by the Waugh Family Chair in Multiple Sclerosis. Karen Tu has received research funding from the Department of Defense United States of America, MaRS Innovation Fund, Canadian Institutes of Health Research Cover Letter (CIHR), Rathlyn Foundation, Canadian Dermatology Foundation, Canadian Rheumatology Association (CRA), Canadian Initiative for Outcomes in Rheumatology Care (CIORA), Toronto Rehab Institute Chair Fund, University of Toronto Practice-Based Research Network (UTOPIAN), PSI Foundation, Cancer Care Ontario (CCO) Clinical Programs & Quality Initiatives, Arthritis Society, McLaughlin Centre Accelerator Grant, MS Society of Canada and Consortium of MS Centers, Ontario SPOR Support Unit, Ontario Institute for Cancer Research (OICR) Health Services Research Program, Vascular Network and Heart and Stroke Foundation of Ontario. She also receives a research scholar award from the Department of Family and Community Medicine at the University of Toronto. Susan Schultz has nothing to disclose. Kinwah Fung has nothing to disclose. Colleen Maxwell has received research funding from the Canadian Institutes of Health Research, P.S.I. Foundation, MS Society of Canada, Consortium of MS Centers, MS Scientific Research Foundation, Ontario Ministry of Health and Long-Term Care-Health System Research Fund, Canadian Frailty Network, Partners for Canadian Consortium on Neurodegeneration in Aging, Network for Aging Research-University of Waterloo. She is supported by a University Research Chair at the University of Waterloo. This does not alter our adherence to PLOS ONE policies on sharing data and materials.

conducted this study in Ontario, Canada, a region that provides an ideal environment for isolating the effect of immigrant status beyond health coverage as it has one of the largest and most diverse immigrant populations globally [14]; and because residents of Ontario have access to universal health insurance without user fees. In a previous large, community-based study, immigrants reported less favourable access to health services compared to native-born individuals in both Canada and the United States, although immigrants in Canada fared relatively better than their American counterparts [1].

We aimed to compare health services utilization in immigrants versus long-term residents with incident multiple sclerosis (MS) in Ontario, Canada. We focused on utilization before, during, and after the MS diagnostic year. Identifying disparities in health service utilization in immigrants can lead to better understanding of inequities in health care outcomes and strategies to mitigate these disparities.

## Methods

### Setting

This retrospective cohort study was performed in the province of Ontario, Canada, using linked health administrative data. At more than 14 million residents, Ontario is the province with the largest population in Canada, and is home to the largest contingent of immigrants in Canada [15]. Billings and administrative data are collected through OHIP (the Ontario Health Insurance Plan), which is a universal, publicly funded program that covers both in-patient and out-patient physician and hospital services without user fees.

### Data sources

The following datasets were linked by unique encoded identifiers and analyzed at ICES. The National Ambulatory Care Reporting System (NACRS) in Ontario captures emergency department (ED) visits, including visit date and the reason for visit, which is reported using International Classification of Disease 10th revision Canadian enhancement (ICD 10-CA) codes. The Canadian Institute for Health Information Discharge Abstract Database (CIHI-- DAD) provides data on hospitalizations including dates of admission and discharge as well as diagnoses recorded using ICD-10-CA codes. We obtained demographic information from the Registered Persons Database (RPDB) of Ontario, including age, sex, health insurance eligibility, postal code of residence (used to derive neighbourhood income quintile and urban/rural residence), and death information. Outpatient physician visits were determined from physician billing claims from OHIP. Immigrant data were obtained through the electronic Permanent Resident Database of Immigration, Refugees and Citizenship Canada (IRCC), as detailed below.

### Study populations

**Immigrants.** The electronic IRCC data captures all immigrants arriving in Canada since 1985. Immigrant data have been linked with OHIP data using probabilistic matching [14] and with the RPDB. We thus identified immigrants arriving in Ontario since 1985. Long-term residents were defined as individuals born in Ontario or whose first date of residence in Ontario occurred before 1985. The same approach was used in several previous studies of immigrants in Ontario [14,16].

**MS cases.** Cases of MS were identified using a validated algorithm applied to health administrative billing data [17]. Briefly, the algorithm required one hospitalization or at least 5 outpatient visits for MS over a two-year period using the International Classification of Disease

(ICD)-9/10-CA codes 340/G35. This algorithm has a sensitivity of approximately 85% and a specificity approaching 100% [17]. The index date was determined by the first in-patient or out-patient contact for a demyelinating condition as indicated by ICD-9/10-CA codes, including encephalomyelitis (323/G36 or G37), optic neuritis (377/H46), or MS (340/G35), in individuals who later met the MS algorithm. For hospitalizations, both primary and secondary discharge diagnoses of MS (340/G35) were included.

The year of MS diagnosis was defined as the period from six months before until 6 months after the index date (Fig 1), to account for symptoms arising weeks to months in advance of the diagnostic visit. A similar approach has been used in previous studies [18,19]. Hence, the year before MS diagnosis was defined as the year up until 6 months before the index date, and the first year following diagnosis began 6 months after the index date. The second year following diagnosis was thus defined from 1.5 years up to 2.5 years following the index date.

## Cohort selection

Cohort selection is shown in Fig 2. Briefly, data collection began in 2003 because this was the first year when electronic data were available for all datasets. Incident MS cases were identified between 1 January 2003 and 31 December 2014. End of study for health service utilization data was up until 30 June 2017 to allow for 2.5 years of follow-up after the MS index date. OHIP eligibility of at least 2 years was required for cohort selection to ensure that we were identifying incident as opposed to prevalent cases. We included those who were ≥20 or ≤65 years of age at cohort entry. The lower limit on age was imposed because MS is rare in children and the algorithm for identifying MS cases in Ontario had not been validated for those <20 years at the time this study started [20]. The upper limit on age was chosen because of the relative rarity of new MS diagnoses and possible misdiagnosis at age >65 years [21]. Individuals who were not resident in Ontario or those who died at the time of MS diagnosis were excluded.

## Outcomes

The number of hospital admissions per person were recorded for year before MS diagnosis, year of diagnosis, and first two years following diagnosis. The number of outpatient neurology, other specialist (e.g. surgery, obstetrics, internal medicine, psychiatry, ophthalmology) and primary care visits per year were likewise recorded. We determined whether MS (340/G35) was listed as the primary diagnosis or cause of the visit for all years except for the year before MS diagnosis and for all visit types except for outpatient neurology visits, because of the potential for conflation of different indications for neurologic consultation.

## Covariates

Covariates included MS diagnosis year (grouped as 2003–04 (reference group), 2005–09, and 2010–14), age group at MS index date (age 20–35, 36–50, and 51–65 years; the latter was the

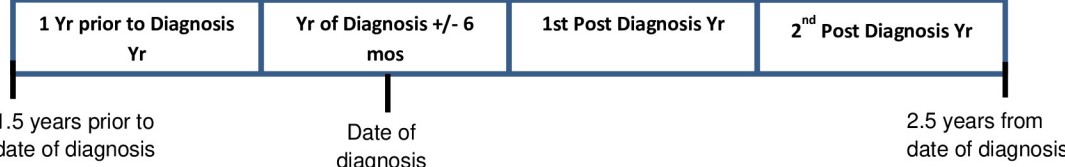

**Fig 1. Health service utilization periods relative to MS diagnosis.** Fig 1 demonstrates the four periods of health service utilization relative to the date of MS diagnosis.

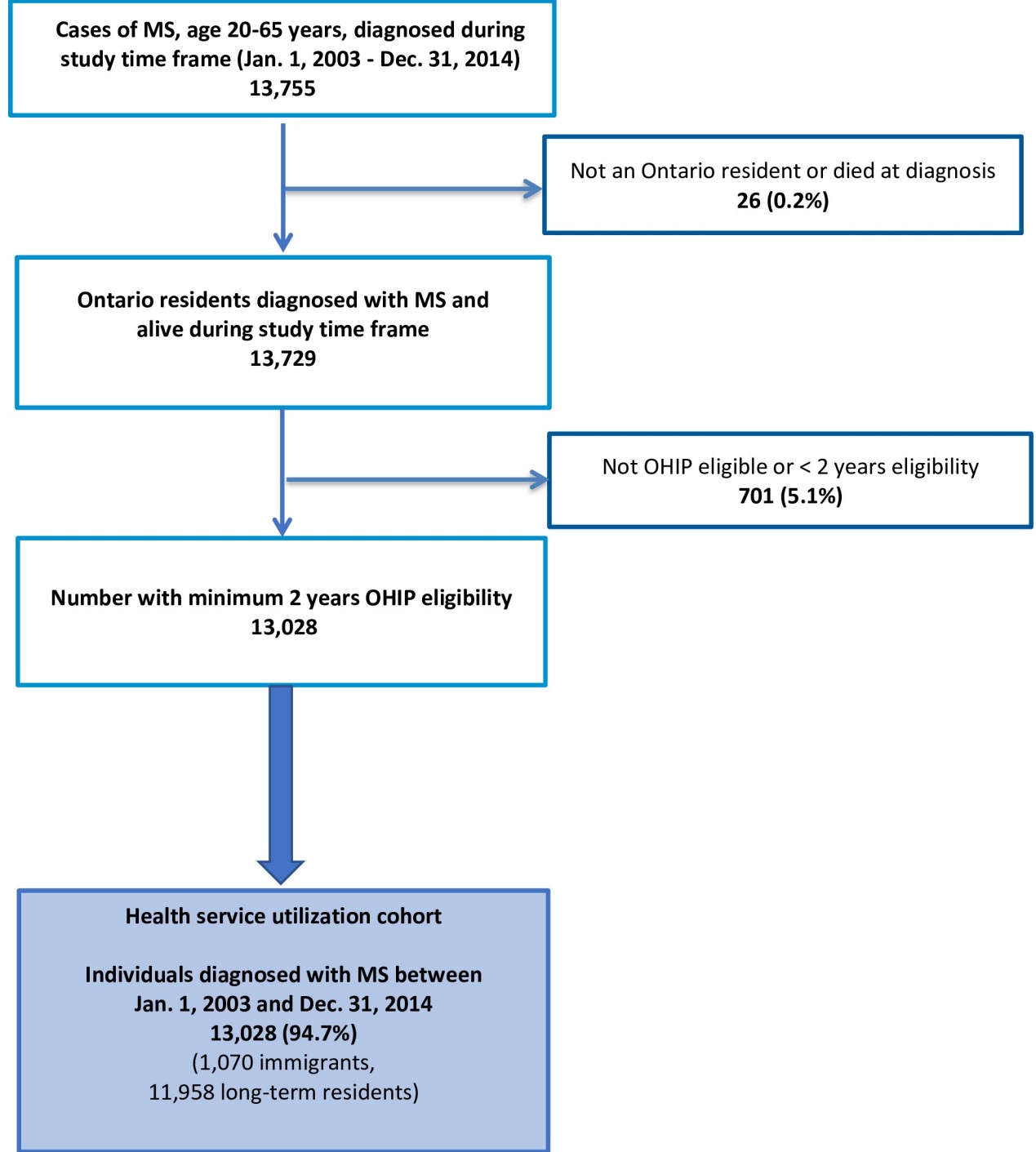

**Fig 2. Health service utilization cohort selection flowchart.** Fig 2 illustrates selection of the study cohort of individuals diagnosed with MS.

reference group), sex, neighbourhood income quintile, urban/rural residence, and comorbidity burden. Covariates were defined as previously described [22]. Briefly, urban residence was classified by residence in any area with a population of ≥10,000 residents. Postal code of residence was linked with the nearest year of Canadian Census data to determine urban residence

and neighbourhood income quintile. Comorbidity burden was measured using ACG® System Aggregated Diagnosis Groups (ADGs) Mortality Risk Score, a weighted score which incorporates ADGs and predicts 1-year mortality [23]. ADGs are based on the Johns Hopkins ACG® System Ver. 10, and are determined from hospitalization and physician visit data for the two years before the index date. Region of origin was included in the descriptive baseline characteristics of the immigrant cohort and classified as Africa, the Caribbean, East Asia, Hispanic America, the Middle East, South Asia, and Western countries (the United States, Europe, Russia, and Australia).

## Analysis

Baseline characteristics of immigrants versus long-term residents at MS index date were summarized using descriptive statistics. We compared the cohorts using Student's t-tests for continuous variables and chi-square tests for categorical variables. Crude rates of number of visits per 100 person-years (with 95% confidence intervals [CI]) were calculated for hospitalizations, ED visits, outpatient neurology and outpatient primary care visits based on a negative binomial distribution. Hospital length of stay was reported in days. For hospital admissions and ED visits, we analyzed MS-specific and total number of visits separately. MS-specific visits were not considered in the year before diagnosis because by our cohort definitions there were no recorded MS indications at that time. We compared utilization rates between immigrants and long-term residents using rate ratios (RR) and 95% CI.

We then evaluated the association between immigrant status and health services utilization using negative binomial regression models, fitted via generalized estimating equations with an exchangeable correlation matrix to account for dependencies introduced by multiple measurements per individual. To account for partial residency time during a given year, we included the logarithm of person-time as an offset in all models. To identify yearly trends, an interaction term was included between immigrant status (yes or no) and the period (-1 to 2; i.e. –1 indicates 1 year before the diagnosis year) in all models. Rate ratios were adjusted for age at MS diagnosis, sex, income quintile, comorbidity burden (weighted ADG score), MS diagnosis calendar year, and urban/rural residence using the negative binomial regression models.

Model assumptions were tested using standard methods [24]. Statistical analyses were conducted using SAS version 9.4 (SAS Institute Inc., Cary, NC).

This study was approved by the research ethics boards at St. Michael's Hospital in Toronto, Ontario, Canada. Use of Ontario data in this study was authorized under section 45 of Ontario's Personal Health Information Protection Act. Section 45 authorizes ICES to collect personal health information, without consent, for the purpose of analysis or compiling statistical information with respect to the management of, evaluation or monitoring of, the allocation of resources to or planning for all or part of the health system.

## Results

We identified 13,755 incident MS cases in Ontario from 2003–2014; 13,028 (94.7%) qualified for inclusion (Fig 2). Among the analytical sample, 1,070 (8.2%) were immigrants. Immigrants were slightly younger at diagnosis, had overall lower socioeconomic status as represented by neighbourhood income quintile, more often lived in urban areas, had lower mean comorbidity burden, and were more likely to enter the cohort in the later years of the study (Table 1). More than 81% had been living in Canada for at least 5 years. There was no significant difference observed in sex distribution across the groups. The most highly represented regions of origin were Western countries (including Europe and North America) and the Middle East.

**Table 1. Baseline characteristics of cohort comparing immigrants versus long-term residents with MS.**

| Variable | Value | Immigrants N = 1,070 | Long-term residents N = 11,958 | Total N = 13,028 | P-VALUE |
|---|---|---|---|---|---|
| **Diagnosis Year** | 2003 | 64 (6.0%) | 987 (8.3%) | 1,051 (8.1%) | <0.001 |
| | 2004 | 66 (6.2%) | 1,022 (8.5%) | 1,088 (8.4%) | |
| | 2005 | 71 (6.6%) | 1,047 (8.8%) | 1,118 (8.6%) | |
| | 2006 | 88 (8.2%) | 984 (8.2%) | 1,072 (8.2%) | |
| | 2007 | 84 (7.9%) | 916 (7.7%) | 1,000 (7.7%) | |
| | 2008 | 83 (7.8%) | 957 (8.0%) | 1,040 (8.0%) | |
| | 2009 | 103 (9.6%) | 1,367 (11.4%) | 1,470 (11.3%) | |
| | 2010 | 95 (8.9%) | 1,296 (10.8%) | 1,391 (10.7%) | |
| | 2011 | 94 (8.8%) | 832 (7.0%) | 926 (7.1%) | |
| | 2012 | 107 (10.0%) | 862 (7.2%) | 969 (7.4%) | |
| | 2013 | 109 (10.2%) | 808 (6.8%) | 917 (7.0%) | |
| | 2014 | 106 (9.9%) | 880 (7.4%) | 986 (7.6%) | |
| **Age at MS index, years** | Mean ± SD | 38.17 ± 10.56 | 39.72 ± 11.20 | 39.59 ± 11.15 | <0.001 |
| **Age at MS Index, years (groups)** | 20–35 | 452 (42.2%) | 4,699 (39.3%) | 5,151 (39.5%) | <0.001 |
| | 36–50 | 479 (44.8%) | 4,970 (41.6%) | 5,449 (41.8%) | |
| | 51–65 | 139 (13.0%) | 2,289 (19.1%) | 2,428 (18.6%) | |
| **Sex** | Female | 742 (69.3%) | 8,093 (67.7%) | 8,835 (67.8%) | 0.263 |
| | Male | 328 (30.7%) | 3,865 (32.3%) | 4,193 (32.2%) | |
| **Neighbourhood income quintile** | 1—lowest | 277 (25.9%) | 2,241 (18.7%) | 2,518 (19.3%) | <0.001 |
| | 2 | 184 (17.2%) | 2,295 (19.2%) | 2,479 (19.0%) | |
| | 3 | 214 (20.0%) | 2,405 (20.1%) | 2,619 (20.1%) | |
| | 4 | 244 (22.8%) | 2,569 (21.5%) | 2,813 (21.6%) | |
| | 5—highest | 150 (14.0%) | 2,383 (19.9%) | 2,533 (19.4%) | |
| **Rural residence** | Urban | 1,047 (97.9%) | 10,372 (86.7%) | 11,419 (87.6%) | <0.001 |
| | Rural | 23 (2.1%) | 1,582 (13.2%) | 1,605 (12.3%) | |
| **Weighted ADG score** | Mean ± SD | 7.40 ± 11.63 | 8.42 ± 11.84 | 8.34 ± 11.82 | 0.007 |
| | Median (IQR) | 5 (1–14) | 6 (1–16) | 6 (1–15) | 0.007 |
| **Index year** | 2003–2004 | 130 (12.1%) | 2,009 (16.8%) | 2,139 (16.4%) | <0.001 |
| | 2005–2009 | 429 (40.1%) | 5,271 (44.1%) | 5,700 (43.8%) | |
| | 2010–2014 | 511 (47.8%) | 4,678 (39.1%) | 5,189 (39.8%) | |
| **Reason for exit** | Out migration/loss of OHIP eligibility | 14 (1.3%) | 137 (1.1%) | 151 (1.2%) | 0.029 |
| | Death | 46 (4.3%) | 755 (6.3%) | 801 (6.1%) | |
| | December 31, 2016 | 1,010 (94.4%) | 11,066 (92.5%) | 12,076 (92.7%) | |
| **Follow-up years since MS diagnosis** | Mean ± SD | 7.17 ± 3.46 | 7.80 ± 3.47 | 7.75 ± 3.48 | <0.001 |
| **Years since landing** | Long Term Residents | 0 (0.0%) | 11,958 (100.0%) | 11,958 (91.8%) | <0.001 |
| | within 2 | 36 (3.4%) | 0 (0.0%) | 36 (0.3%) | |
| | 2–5 | 165 (15.4%) | 0 (0.0%) | 165 (1.3%) | |
| | 5–10 | 269 (25.1%) | 0 (0.0%) | 269 (2.1%) | |
| | 10–15 | 228 (21.3%) | 0 (0.0%) | 228 (1.8%) | |
| | 15+ | 372 (34.8%) | 0 (0.0%) | 372 (2.9%) | |
| **Region of origin** | Africa | 46 (4.3%) | 0 (0.0%) | 46 (0.4%) | |
| | Caribbean | 80 (7.5%) | 0 (0.0%) | 80 (0.6%) | |
| | East Asia | 74 (6.9%) | 0 (0.0%) | 74 (0.6%) | |
| | Hispanic America | 64 (6.0%) | 0 (0.0%) | 64 (0.5%) | |
| | Middle East | 249 (23.3%) | 0 (0.0%) | 249 (1.9%) | |
| | South Asia | 128 (12.0%) | 0 (0.0%) | 128 (1.0%) | |

(*Continued*)

**Table 1.** (Continued)

| Variable | Value | Immigrants N = 1,070 | Long-term residents N = 11,958 | Total N = 13,028 | P-VALUE |
|---|---|---|---|---|---|
| | Western | 392 (36.6%) | 0 (0.0%) | 392 (3.0%) | |

MS = multiple sclerosis.

ADG = Aggregated Diagnosis Groups.

### Health service utilization analysis

**ED visits.** Immigrants had lower rates of total ED use in the year before, the year of diagnosis and for the two post-diagnosis years (Table 2 and Fig 3A). Rates of MS-specific ED visits were similar in immigrants compared to long-term residents in the year of diagnosis and the two years post-diagnosis.

**Hospitalizations.** Rates of hospitalization in immigrants were similar to long-term residents in the year before diagnosis (Table 2). Immigrants had higher rates of both total and MS-specific hospitalizations in the MS diagnosis year (Table 2 and Fig 3B). In contrast, total and MS-specific hospitalizations were less frequent in immigrants in the first post-diagnosis year, but not significantly different from long-term residents in the second post-diagnosis year. Hospital length of stay (days) was longer in immigrants but this difference was not statistically significant.

**Primary care visits.** Primary care visits, both total and MS-specific were less frequent in immigrants from the year before MS diagnosis through two years following diagnosis (Table 2 and Fig 3C).

**Neurology outpatient visits.** Immigrants had lower rates of total neurology outpatient visits in the year before diagnosis, but higher rates of neurology specialty care in the year of MS diagnosis and for the two years post-diagnosis (Table 2 and Fig 3C).

**Other specialist outpatient visits.** Other specialist outpatient visits were more likely among immigrants in the year before MS diagnosis, the year of diagnosis, and the first post-diagnosis year (Table 2). In the second post diagnosis year, they continued to be more frequent in immigrants but this finding did not meet statistical significance. Similarly, MS-specific outpatient visits tended to be more common in immigrants in the year of MS diagnosis through two years following diagnosis but these observations were not statistically significant.

For most variables the regression analysis did not substantially alter the observed relationships. For the year of MS diagnosis, the relationship between immigrant status and total inpatient hospitalizations became statistically significant with adjustment for covariates; the relationship between immigrant status and MS-specific hospitalizations was statistically significant in both the adjusted and unadjusted analyses.

## Discussion

This novel study examined health services utilization in a large, population-based cohort of immigrants and long-term Ontario residents with MS. Immigrants demonstrated a higher rate of hospitalization in the year of MS diagnosis, despite greater use of neurology outpatient care. Other notable findings were a lower rate of primary care outpatient visits in immigrants and higher rates of other, non-neurology outpatient specialist visits. Overall, these findings suggest that immigrants with MS in Ontario, Canada, a region with universal health care insurance, are not systematically disadvantaged in their access to health services. Individuals with MS are expected to have higher rates of health services use compared to the general population, although many MS-related symptoms can be managed today on an outpatient basis [18,25,26].

**Table 2. Unadjusted and adjusted rate ratios comparing health services utilization in immigrants vs. long-term residents with multiple sclerosis.**

| | | Unadjusted | | | Adjusted* | | |
|---|---|---|---|---|---|---|---|
| | Time Period | Rate Ratio | 95% CI | | Rate Ratio | 95% CI | |
| Total ED visits | 1 year before diagnosis year | **0.59** | **0.52** | **0.67** | **0.66** | **0.58** | **0.74** |
| | Diagnosis year | **0.82** | **0.73** | **0.91** | **0.86** | **0.78** | **0.96** |
| | 1st post diagnosis year | **0.61** | **0.54** | **0.69** | **0.65** | **0.58** | **0.74** |
| | 2nd post diagnosis year | **0.68** | **0.60** | **0.77** | **0.72** | **0.64** | **0.81** |
| ED visits for MS | Diagnosis year | 1.06 | 0.79 | 1.42 | 1.07 | 0.80 | 1.43 |
| | 1st post diagnosis year | 0.82 | 0.59 | 1.15 | 0.84 | 0.61 | 1.16 |
| | 2nd post diagnosis year | 0.84 | 0.61 | 1.18 | 0.92 | 0.67 | 1.28 |
| Total inpatient hospitalizations | 1 year before diagnosis year | **0.79** | **0.66** | **0.94** | 0.86 | 0.73 | 1.01 |
| | Diagnosis year | 1.11 | 0.95 | 1.29 | **1.20** | **1.04** | **1.39** |
| | 1st post diagnosis year | **0.74** | **0.62** | **0.88** | **0.79** | **0.67** | **0.94** |
| | 2nd post diagnosis year | 0.87 | 0.73 | 1.04 | 0.95 | 0.80 | 1.12 |
| Inpatient hospitalizations for MS | Diagnosis year | **1.21** | **1.03** | **1.44** | **1.28** | **1.08** | **1.51** |
| | 1st post diagnosis year | **0.73** | **0.60** | **0.89** | **0.77** | **0.63** | **0.94** |
| | 2nd post diagnosis year | 0.93 | 0.75 | 1.14 | 0.97 | 0.79 | 1.19 |
| Inpatient hospitalization length of stay (days) | 1 year before diagnosis year | 1.25 | 0.86 | 1.82 | 1.09 | 0.77 | 1.55 |
| | Diagnosis year | 1.15 | 0.83 | 1.61 | 1.17 | 0.86 | 1.60 |
| | 1st post diagnosis year | 1.06 | 0.73 | 1.54 | 1.28 | 0.91 | 1.82 |
| | 2nd post diagnosis year | 1.04 | 0.72 | 1.51 | 1.08 | 0.76 | 1.53 |
| Inpatient hospitalization for MS, length of stay (days) | Diagnosis year | 1.04 | 0.66 | 1.62 | 1.15 | 0.74 | 1.78 |
| | 1st post diagnosis year | 1.42 | 0.86 | 2.35 | 1.70 | 1.04 | 2.75 |
| | 2nd post diagnosis year | 1.21 | 0.74 | 2.01 | 1.60 | 0.98 | 2.61 |
| Total primary care outpatient visits | 1 year before diagnosis year | **0.81** | **0.74** | **0.88** | **0.88** | **0.82** | **0.95** |
| | Diagnosis year | **0.77** | **0.71** | **0.82** | **0.84** | **0.78** | **0.90** |
| | 1st post diagnosis year | **0.69** | **0.63** | **0.75** | **0.75** | **0.69** | **0.81** |
| | 2nd post diagnosis year | **0.71** | **0.65** | **0.77** | **0.79** | **0.73** | **0.86** |
| Primary care outpatient visits for MS | Diagnosis year | **0.73** | **0.65** | **0.82** | **0.76** | **0.67** | **0.85** |
| | 1st post diagnosis year | **0.79** | **0.70** | **0.91** | **0.81** | **0.71** | **0.92** |
| | 2nd post diagnosis year | **0.85** | **0.74** | **0.97** | **0.87** | **0.76** | **0.99** |
| Neurologist outpatient visits | 1 year before diagnosis year | 0.95 | 0.89 | 1.01 | **0.93** | **0.87** | **0.99** |
| | Diagnosis year | **1.19** | **1.13** | **1.25** | **1.17** | **1.12** | **1.23** |
| | 1st post diagnosis year | **1.18** | **1.12** | **1.25** | **1.16** | **1.10** | **1.23** |
| | 2nd post diagnosis year | **1.18** | **1.11** | **1.24** | **1.16** | **1.09** | **1.22** |
| Total other specialist outpatient visits | 1 year before diagnosis year | **1.21** | **1.08** | **1.36** | **1.22** | **1.09** | **1.36** |
| | Diagnosis year | **1.22** | **1.11** | **1.35** | **1.23** | **1.11** | **1.36** |
| | 1st post diagnosis year | **1.19** | **1.06** | **1.33** | **1.19** | **1.06** | **1.33** |
| | 2nd post diagnosis year | 1.07 | 0.96 | 1.20 | 1.07 | 0.96 | 1.20 |
| Other specialist outpatient visits for MS | Diagnosis year | 1.08 | 0.81 | 1.46 | 1.15 | 0.85 | 1.55 |
| | 1st post diagnosis year | 0.97 | 0.70 | 1.36 | 1.06 | 0.76 | 1.48 |
| | 2nd post diagnosis year | 1.07 | 0.77 | 1.50 | 1.18 | 0.85 | 1.64 |

*Rate ratios were adjusted for age at MS diagnosis, sex, neighbourhood income quintile, comorbidity burden (weighted ADG score), urban/rural residence, and MS diagnosis calendar year using negative binomial regression models.

95% CI = 95% Confidence Interval.

Bold face: statistically significant.

MS = multiple sclerosis.

ED = Emergency Department.

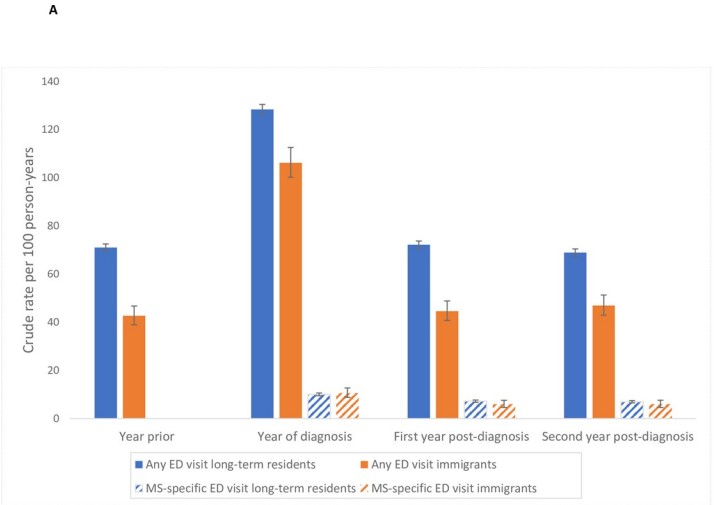

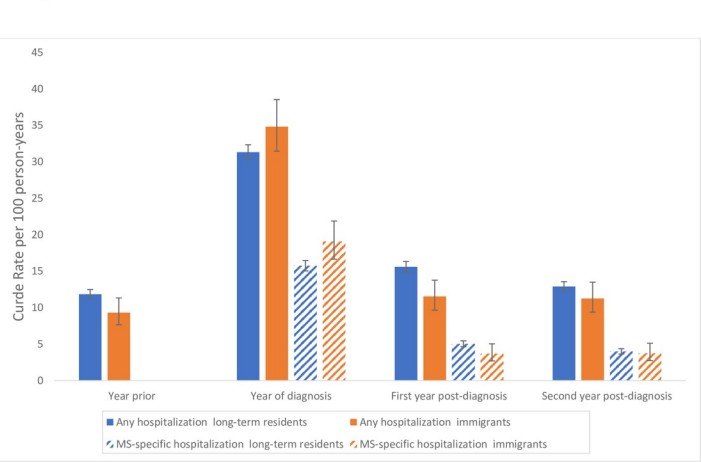

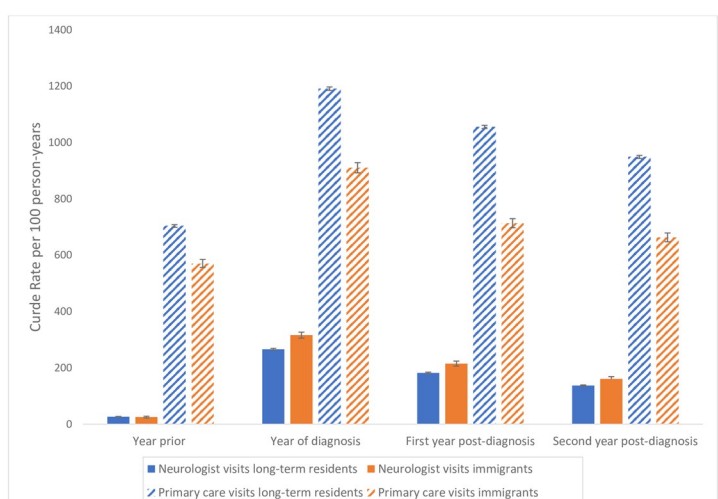

**Fig 3. Crude rates of health service utilization in immigrants vs. long-term residents with MS.** Crude rates of health service utilization are shown for immigrants with MS compared to long-term residents with MS for (A) Emergency Department visits, (B) hospitalizations, and (C) outpatient neurology and primary care visits. For Emergency Department visits and hospitalizations, all event and MS-specific event rates are displayed.

In a previous scoping review of the literature, non-white race was associated with less access to health services in individuals with MS, although immigrant status was not investigated [9]. Studies of health service utilization may improve understanding of disparities across sociodemographic groups and in turn support policies to address those disparities.

However, despite the generally positive findings concerning access to health services in immigrants with MS in Ontario, the finding of a higher rate of hospitalization in the year of MS diagnosis is concerning, as higher use of outpatient neurology care is generally associated with fewer hospitalizations in neurologic diseases [6]. Possible explanations could include a more severe or sudden presentation of neurologic symptoms in immigrants with MS. More severe early demyelinating attacks and less neurologic recovery have been observed in individuals of non-white race in past research [27,28]. We previously reported a higher risk of mortality in immigrants compared to long-term residents in the year following MS onset, although mortality in immigrants was lower thereafter [22]. Alternatively, immigrants may lack health

system literacy and social supports which could mitigate the consequences of a first attack or other illness. However, a previous study found that economic class immigrants to Ontario had lower overall rates of hospitalization than the general population, possibly a reflection of the healthy migrant effect [29]. Immigrants to Ontario had lower rates of hospitalization for circulatory diseases in another study [30]. We did not observe higher rates of total or MS-specific hospitalization in immigrants outside of the MS-diagnosis year, which may argue against social supports playing a prominent role in the hospitalization patterns observed.

Lower rates of use of certain health services including outpatient neurology by immigrants in the year before MS diagnosis also raise the question of whether immigrants may have a shorter or less discernible MS prodrome, with more abrupt onset of demyelinating symptoms, compared to long-term residents in Canada. The MS prodrome refers to the onset of early symptoms that precede presentation with classic symptoms of the disease, as recognized by the observation that health service utilization increases steadily in individuals with MS relative to the general population beginning five years up until the time of MS diagnosis [26,31]. Previously, a study of prodromal symptoms in women who later developed myocardial infarction found that race influenced the extent of prodromal symptoms, although the investigators did not assess immigrant status [32]. We also cannot exclude the possibility that immigrants experience similar symptoms before MS diagnosis as long-term residents but are less likely to seek out care because of financial, cultural, or language barriers and therefore may seek healthcare at later stages of the disease.

Strengths of this study include the availability of population-based continuous health services data, use of a validated algorithm for identifying MS cases [17], and the large cohort of immigrants and long-term residents with MS. We controlled for age, sex, socioeconomic status, urban/rural residence, overall comorbidity burden, and calendar year of diagnosis which may all contribute to disparities in health services use. Limitations of this study include lack of availability of prescription drug data which could have provided a more complete picture of access to needed health care resources among immigrants with MS. We cannot exclude the possibility of systematic differences in DMT prescriptions between immigrants and long-term residents but believe this is unlikely due to government coverage of DMT in Ontario for those without private drug insurance. The validated algorithm used to identify MS cases has a sensitivity of about 85%, and data concerning immigrants did not extend back before 1985, thus some MS cases may have been missed. We could not assess for differences in disability or disease severity between the immigrant and long-term resident populations because of lack of clinical data in the administrative health claims databases. Socioeconomic status was approximated by neighbourhood income quintile because individual-level variables such as household income are not collected as part of the health administrative data in Ontario. There may be some year-to-year stochastic variation in health service use. Last, this study focused on a region with universal health insurance with health services free at the point of care; health services use in immigrants with MS may differ under other insurance models and in other geographic regions.

## Conclusions

In conclusion, immigrants with MS in Ontario, Canada demonstrated a generally favourable pattern of access to health services, with greater use of neurology outpatient care and lower use of emergency department services. However, in the year of MS diagnosis immigrants had a higher rate of both total and MS-specific hospitalizations compared to long-term residents. Reasons for this are unclear but warrant further investigation and may include a more severe or abrupt presentation of neurologic symptoms of MS in immigrants.

## Author Contributions

**Conceptualization:** Ruth Ann Marrie, Karen Tu, Colleen J. Maxwell.

**Formal analysis:** Dalia L. Rotstein, Susan E. Schultz, Kinwah Fung.

**Funding acquisition:** Dalia L. Rotstein, Ruth Ann Marrie, Karen Tu, Colleen J. Maxwell.

**Methodology:** Dalia L. Rotstein, Ruth Ann Marrie, Karen Tu, Colleen J. Maxwell.

**Supervision:** Dalia L. Rotstein, Ruth Ann Marrie, Colleen J. Maxwell.

**Writing – original draft:** Dalia L. Rotstein.

**Writing – review & editing:** Dalia L. Rotstein, Ruth Ann Marrie, Karen Tu, Susan E. Schultz, Kinwah Fung, Colleen J. Maxwell.

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
