## [Decision Letter · Decision Letter 0]

16 Apr 2020

PONE-D-20-05380

Health service utilization in immigrants with multiple sclerosis

PLOS ONE

Dear Dr. Rotstein,

Thank you for submitting your manuscript to PLOS ONE. After careful consideration, we feel that it has merit but does not fully meet PLOS ONE’s publication criteria as it currently stands. Therefore, we invite you to submit a revised version of the manuscript that addresses the points raised during the review process.

We would appreciate receiving your revised manuscript by May 31 2020 11:59PM. To enhance the reproducibility of your results, we recommend that if applicable you deposit your laboratory protocols in protocols.io, where a protocol can be assigned its own identifier (DOI) such that it can be cited independently in the future. For instructions see: http://journals.plos.org/plosone/s/submission-guidelines#loc-laboratory-protocols

We look forward to receiving your revised manuscript.

Kind regards,

Astrid M. Kamperman

Academic Editor

PLOS ONE

Journal Requirements:

2. Thank you for stating the following in the Financial Disclosure section:"This study was funded through grants from the Multiple Sclerosis Society of Canada and the Consortium of Multiple Sclerosis Centers.  The funding sources had no role in study design, data collection, data analysis, data interpretation, or writing of the report. All authors had full access to the data in the study and the corresponding author had the final responsibility for the decision to submit for publication. The sponsors played no role in the study design; in the collection, analysis and interpretation of data; in the writing of the report; or in the decision to submit the report for publication."

Thank you for stating the following in the Competing Interests section: " I have read the journal's policy and the authors of this manuscript have the following competing interests:

Dalia Rotstein has received research support from the Consortium of Multiple Sclerosis Centers (CMSC), the Multiple Sclerosis Society of Canada, and Roche Canada. She has served as a speaker or consultant for Alexion, Biogen, EMD Serono, Novartis, Roche and Sanofi Aventis.

Ruth Ann Marrie receives research funding from: CIHR, Research Manitoba, Multiple Sclerosis Society of Canada, Multiple Sclerosis Scientific Foundation, Crohn’s and Colitis Canada, National Multiple Sclerosis Society, CMSC and United States Department of Defence. She is supported by the Waugh Family Chair in Multiple Sclerosis.

Karen Tu has received research funding from the Department of Defense United States of America, MaRS Innovation Fund, Canadian Institutes of Health Research (CIHR), Rathlyn Foundation, Canadian Dermatology Foundation, Canadian Rheumatology Association (CRA), Canadian Initiative for Outcomes in Rheumatology Care (CIORA), Toronto Rehab Institute Chair Fund, University of Toronto Practice-Based Research Network (UTOPIAN), PSI Foundation, Cancer Care Ontario (CCO) Clinical Programs & Quality Initiatives, Arthritis Society, McLaughlin Centre Accelerator Grant, MS Society of Canada and Consortium of MS Centers, Ontario SPOR Support Unit, Ontario Institute for Cancer Research (OICR) Health Services Research Program, Vascular Network and Heart and Stroke Foundation of Ontario. She also receives a research scholar award from the Department of Family and Community Medicine at the University of Toronto.

Susan Schultz has nothing to disclose.

Kinwah Fung has nothing to disclose.

Colleen Maxwell has received research funding from the Canadian Institutes of Health Research, P.S.I. Foundation, MS Society of Canada, Consortium of MS Centers, MS Scientific Research Foundation, Ontario Ministry of Health and Long-Term Care-Health System Research Fund, Canadian Frailty Network, Partners for Canadian Consortium on Neurodegeneration in Aging, Network for Aging Research-University of Waterloo. She is supported by a University Research Chair at the University of Waterloo. "

We note that one or more of the authors are employed by a commercial company: H2 Toronto

b) . Please also provide an updated Competing Interests Statement declaring this commercial affiliation along with any other relevant declarations relating to employment, consultancy, patents, products in development, or marketed products, etc. 

Reviewers' comments:

Reviewer's Responses to Questions

**Comments to the Author**

1. Is the manuscript technically sound, and do the data support the conclusions?

Reviewer #1: Yes

Reviewer #2: Yes

2. Has the statistical analysis been performed appropriately and rigorously? 

Reviewer #1: Yes

Reviewer #2: Yes

3. Have the authors made all data underlying the findings in their manuscript fully available?

Reviewer #1: Yes

Reviewer #2: Yes

4. Is the manuscript presented in an intelligible fashion and written in standard English?

Reviewer #1: Yes

Reviewer #2: Yes

5. Review Comments to the Author

Reviewer #1: The authors have addressed the important topic of disparities in health services utilization for MS between immigrants and non-immigrants, for which there is a dearth of literature. By focusing on a publicly funded healthcare system in which residents have access to universal health insurance without user fees, the authors are able to focus on disparities that persist despite barriers associated with insurance coverage and payment. The manuscript is well-written. The methods and analyses are sound and are well-described and the conclusions are warranted. My main concerns focus on the framing of the manuscript, which right now lacks context and does not provide a sufficient framework for systematically understanding immigrant disparities in health services utilization for MS.

The first sentence in the introduction states that “disparity in health service utilization across sociodemographic groups is an important but poorly studied phenomenon which may influence health outcomes.” This statement does not accurately characterize the disparities literature, as there have been many high-quality health services research studies documenting the presence and correlates of healthcare disparities in utilization across many disease conditions. The author should restructure the introduction to focus more narrowly on the specific gap in the literature that this study addresses: namely examining disparities in health services utilization for neurological diseases (MS, specifically) between immigrants and non-immigrants.

A second issue is that the authors do not discuss, in the introduction, what type(s) of utilization patterns would constitute a disparity indicative of worse MS care. This makes it difficult to understand how to interpret the results. Typically, in disparity research it is clearly stated what would constitute a disparity (e.g., lower rates of kidney transplantation, lower rates of having a certain drug prescribed). The authors should explicitly state this in the introduction, since many PLOS One readers will not be familiar with patterns of utilization associated with high quality care for MS, especially since they examine 5 types of care (hospitalizations, emergency department (ED) visits, outpatient neurology visits, other outpatient specialty visits, and primary care visits) at 3 different time points. It would be helpful to know how one should interpret the meaning of potential disparities for these 5 outcomes overall, and whether this meaning changes based on whether they occurred in the year before diagnosis, the year of diagnosis, and the two years following diagnosis. For example, are higher rates of neurological visits during the post-diagnosis period different, in terms of quality, than higher rates of neurological visits during the pre-diagnosis year? Perhaps a table could be helpful. In the conclusion, the authors state that their findings suggest that immigrants with MS in Ontario, demonstrated “a generally favourable pattern of access to health services with greater use of neurology outpatient care and lower use of emergency department services.” Providing a description, in the introduction, of what a favorable pattern of health services utilization for MS consists of, would make it easier for the reader to interpret the results later and understand how the authors arrived at their conclusions. Along those lines, it would also be helpful to provide an example or examples of what one might learn from these analyses, in terms of how they might improve our understanding of disparities.

A third issue is the lack of a framework, in the introduction, systematically discussing factors that might contribute to disparities between immigrants and non-immigrants. The authors could draw on existing frameworks for presenting this information. This would provide a context for some of their findings that they discuss (such as to why immigrants experience a higher rate of hospitalization in the year of MS diagnosis despite higher use of outpatient neurology). A related point is that, when setting up their study in the introduction, the authors discuss research on racial/ethnic disparities and immigrant disparities without systematically discussing why these two groups’ experiences might differ. It would strengthen the manuscript if the authors discussed specific factors that may contribute to different patterns of disparities in neurological care and MS care for immigrants compared to members of racial and ethnic minority groups.

Related to this, the authors should also be more explicit about why it is useful to focus on disparities within a system of universal health insurance. That is, within such a system, what other factors would be expected to contribute to disparities, for immigrants in general, and for their particular population in Ontario? Along those lines, when presenting their literature review, it would be useful to know whether any of the studies they cite were conducted in systems of universal health insurance.

Minor concern: The authors should replace the term “Caucasian” with white, as the term Caucasian has a problematic history. See Mukhopadhyay, C. C. (2016). Getting rid of the word" Caucasian. Privilege: A Reader, 21.

Reviewer #2: The authors of the manuscript “Health service utilization in immigrants with multiple sclerosis” have written an interesting paper comparing the use of health services in immigrants to Ontario, Canada, compared to long-term residents.

The objectives are clearly defined, and the methods are fitted to answer the research questions. The pros and cons of the paper are mostly satisfactorily discussed. The health system in Canada ensures that the access to health services are universal and without user fees reducing this possible confounder to a minimum. There are however some issues that must be corrected and answered before the paper can be accepted.

1. The socioeconomic status (SES) is roughly defined as the neighbourhood income quintile based on information from the Registered Persons Database of Ontario. SES could be more widely defined as the persons household income, education level, cultural aspects etc. Would it be possible to retrieve more detailed data on the individuals SES? If not, this should be mentioned in the discussion section.

2. The immigrants were identified by electronic IRCC data which captures those arriving in Ontario since 1985. Thus people arriving to Canada from foreign countries before 1985 and later developing MS, would be defined as “long-term residents.” The authors should discuss this issue and eventually how this might affect the results.

3. The first sentence under “Results” refers to Figure 1. This should be corrected to Figure 2.

4. In figure 3, crude rates are given for the health service utilization in immigrants vs. long-term residents, whereas both crude rates and adjusted rates are given in table 2. It should be more clear how the results were affected by these adjustments and if this would alter the conclusions.

5. As the authors mention in the Discussion section, information about prescription of MS drugs is lacking. Different drugs have different side effect and complication rates, including severe reactions that could require hospital admissions. Could possible systematically differences in the prescription of MS drugs between the ethnic groups explain some of the skewness between outpatient visits and hospitalizations?

6. PLOS authors have the option to publish the peer review history of their article (what does this mean?). If published, this will include your full peer review and any attached files.

Reviewer #1: No

Reviewer #2: No

---

## [Author Response · Author response to Decision Letter 0]

22 May 2020

Dear Reviewers,

Thank you for your time, careful review, and thoughtful comments. Please see our responses below. We believe the manuscript has been strengthened with these changes and hope you find it suitable for publication.

Reviewer #1: 

1. My main concerns focus on the framing of the manuscript, which right now lacks context and does not provide a sufficient framework for systematically understanding immigrant disparities in health services utilization for MS.

Response: We agree that the manuscript would benefit from more detailed framing at the outset and have revised the introduction extensively. Please see details below with respect to specific concerns. 

2. The first sentence in the introduction states that “disparity in health service utilization across sociodemographic groups is an important but poorly studied phenomenon which may influence health outcomes.” This statement does not accurately characterize the disparities literature, as there have been many high-quality health services research studies documenting the presence and correlates of healthcare disparities in utilization across many disease conditions. The author should restructure the introduction to focus more narrowly on the specific gap in the literature that this study addresses: namely examining disparities in health services utilization for neurological diseases (MS, specifically) between immigrants and non-immigrants.

Response: Yes, this is an important point. There is a broader literature characterizing disparities in other disease states, which we now mention, then foreground the question regarding how disparities may affect MS care:

Health service access and utilization are known to vary substantially with sociodemographic factors such as sex, race, immigrant status, and income level (1-4). Disparities in health service utilization are reflected in divergent experiences of the health care system and outcomes (1, 5). However, health service utilization patterns have been less extensively described in the context of neurologic diseases, although one American study found that blacks were more likely to be hospitalized, and less likely to experience an ambulatory neurology visit, than whites (6). Health service utilization patterns may be particularly relevant for multiple sclerosis (MS), a chronic inflammatory disease of the central nervous system (CNS) and one of the primary causes of disability in young adults (7). The complex treatment landscape for MS today requires specialized neurology care and early implementation of disease-modifying therapy (DMT) to promote better long-term outcomes (7, 8). Little is known about which sociodemographic factors predict favourable health service utilization in people living with multiple sclerosis (MS).

3. A second issue is that the authors do not discuss, in the introduction, what type(s) of utilization patterns would constitute a disparity indicative of worse MS care. This makes it difficult to understand how to interpret the results. Typically, in disparity research it is clearly stated what would constitute a disparity (e.g., lower rates of kidney transplantation, lower rates of having a certain drug prescribed). The authors should explicitly state this in the introduction, since many PLOS One readers will not be familiar with patterns of utilization associated with high quality care for MS, especially since they examine 5 types of care (hospitalizations, emergency department (ED) visits, outpatient neurology visits, other outpatient specialty visits, and primary care visits) at 3 different time points. It would be helpful to know how one should interpret the meaning of potential disparities for these 5 outcomes overall, and whether this meaning changes based on whether they occurred in the year before diagnosis, the year of diagnosis, and the two years following diagnosis. For example, are higher rates of neurological visits during the post-diagnosis period different, in terms of quality, than higher rates of neurological visits during the pre-diagnosis year? Perhaps a table could be helpful. In the conclusion, the authors state that their findings suggest that immigrants with MS in Ontario, demonstrated “a generally favourable pattern of access to health services with greater use of neurology outpatient care and lower use of emergency department services.” Providing a description, in the introduction, of what a favorable pattern of health services utilization for MS consists of, would make it easier for the reader to interpret the results later and understand how the authors arrived at their conclusions. Along those lines, it would also be helpful to provide an example or examples of what one might learn from these analyses, in terms of how they might improve our understanding of disparities.

Response: Thank you for this suggestion. We agree that it would be helpful to the reader to state in the introduction what constitutes a favourable pattern of health service use with respect to MS. We have added the following paragraph to the introduction:

A favourable pattern of health system utilization in those with established MS diagnoses would be characterized by regular outpatient neurology visits, as comprehensive MS care requires periodic monitoring of the patient’s symptoms, neurologic examination, MRI, and response to DMT. Most MS attacks can be managed on an outpatient basis when there is consistent access to a neurologist; hence fewer emergency department visits and hospitalizations would be expected under favourable conditions (10). However, the initial symptoms or attack in MS may involve an emergency department visit to facilitate diagnostic work-up and possible treatment with corticosteroids (11). Hospitalization is usually required only for more severe attacks (10), the risk of which is reduced with DMT. Regular primary care visits may be appropriate if they complement rather than substitute for specialty neurology care and should be interpreted in that context (12). 

4. A third issue is the lack of a framework, in the introduction, systematically discussing factors that might contribute to disparities between immigrants and non-immigrants. The authors could draw on existing frameworks for presenting this information. This would provide a context for some of their findings that they discuss (such as to why immigrants experience a higher rate of hospitalization in the year of MS diagnosis despite higher use of outpatient neurology). A related point is that, when setting up their study in the introduction, the authors discuss research on racial/ethnic disparities and immigrant disparities without systematically discussing why these two groups’ experiences might differ. It would strengthen the manuscript if the authors discussed specific factors that may contribute to different patterns of disparities in neurological care and MS care for immigrants compared to members of racial and ethnic minority groups.

Response: Thank you for this suggestion. We have added context to distinguish between the effects of race and immigrant status on health service utilization. We have added the following to the introduction:

There has been no previous study of the effect of immigrant status on health service utilization among people with MS, which is the focus of this study. The effect of immigrant status on health service utilization is distinct from the question of race, as health system literacy, language, and cultural factors affect immigrants’ interaction with the health care system (13).

In the Discussion, we mention some other reasons why immigrants may have different patterns of health service use, including a lack of social supports and differences in the manner in which MS may present in different populations:

Possible explanations could include a more severe or sudden presentation of neurologic symptoms in immigrants with MS. More severe early demyelinating attacks and less neurologic recovery have been observed in individuals of non-white race in past research (27, 28). We previously reported a higher risk of mortality in immigrants compared to long-term residents in the year following MS onset, although mortality in immigrants was lower thereafter (22). Alternatively, immigrants may lack healthy system literacy and social supports which could mitigate the consequences of a first attack or other illness.

5. Related to this, the authors should also be more explicit about why it is useful to focus on disparities within a system of universal health insurance. That is, within such a system, what other factors would be expected to contribute to disparities, for immigrants in general, and for their particular population in Ontario? Along those lines, when presenting their literature review, it would be useful to know whether any of the studies they cite were conducted in systems of universal health insurance.

Response: In the following excerpt from the introduction, we outline why it is important to study this question in a universal health insurance system (e.g. to remove the effect of differences in health insurance access in immigrant and non-immigrant population). In the section we mention above, we have added reference to other factors that contribute to disparities in immigrants, namely health system literacy, language, and cultural barriers, with a reference. We have added a reference to a previous study that compared health service utilization in a private health insurance system (the United States) vs. universal health insurance system (Canada) and found challenges for immigrants in both systems, although to a lesser extent in Canada’s universal health insurance system:

We conducted this study in Ontario, Canada, a region that provides an ideal environment for isolating the effect of immigrant status beyond health coverage as it has one of the largest and most diverse immigrant populations globally (14); and because residents of Ontario have access to universal health insurance without user fees. In a previous large, community-based study, immigrants reported less favourable access to health services compared to native-born individuals in both Canada and the United States, although immigrants in Canada fared relatively better than their American counterparts (1). 

Minor concern: The authors should replace the term “Caucasian” with white, as the term Caucasian has a problematic history. See Mukhopadhyay, C. C. (2016). Getting rid of the word" Caucasian. Privilege: A Reader, 21.

Response: Thank you for this suggestion. We have replaced “Caucasian” with “white.” 

Reviewer #2: The authors of the manuscript “Health service utilization in immigrants with multiple sclerosis” have written an interesting paper comparing the use of health services in immigrants to Ontario, Canada, compared to long-term residents.

The objectives are clearly defined, and the methods are fitted to answer the research questions. The pros and cons of the paper are mostly satisfactorily discussed. The health system in Canada ensures that the access to health services are universal and without user fees reducing this possible confounder to a minimum. There are however some issues that must be corrected and answered before the paper can be accepted.

1. The socioeconomic status (SES) is roughly defined as the neighbourhood income quintile based on information from the Registered Persons Database of Ontario. SES could be more widely defined as the persons household income, education level, cultural aspects etc. Would it be possible to retrieve more detailed data on the individuals SES? If not, this should be mentioned in the discussion section.

Response: Thank you for this comment. As mentioned, neighbourhood income quintile was used as a proxy for SES, as has been done in other studies (Tu et al. 2015, in Reference List). Unfortunately it is not possible to derive person-specific socioeconomic measures with the linked administrative health data available for our study. We have added this as a limitation in the Discussion:

Socioeconomic status was approximated by neighbourhood income quintile because individual-level variables such as household income are not collected as part of the health administrative data in Ontario.

2. The immigrants were identified by electronic IRCC data which captures those arriving in Ontario since 1985. Thus people arriving to Canada from foreign countries before 1985 and later developing MS, would be defined as “long-term residents.” The authors should discuss this issue and eventually how this might affect the results.

Response: It is a potential limitation that the data concerning immigrants did not extend back before 1985. However, we do not expect to have missed many cases of MS in immigrants during our study period from 2003-2014 given that the usual age of MS onset is in young adulthood. The same approach has been taken in other studies (Yarnell et al. JAMA 2017, Tu et al. Circulation 2015). We have added mention of this limitation to the Discussion. 

3. The first sentence under “Results” refers to Figure 1. This should be corrected to Figure 2.

Response: Thank you – this has been corrected.

4. In figure 3, crude rates are given for the health service utilization in immigrants vs. long-term residents, whereas both crude rates and adjusted rates are given in table 2. It should be more clear how the results were affected by these adjustments and if this would alter the conclusions.

Response: The results were not substantially altered by the regression analysis with one notable exception. We have added the following to the end of the Results section:

For most variables the regression analysis did not substantially change the observed relationships. For the year of MS diagnosis, the relationship between immigrant status and total inpatient hospitalizations became statistically significant with adjustment for covariates; the relationship between immigrant status and MS-specific hospitalizations was statistically significant in both the adjusted and unadjusted analyses. 

5. As the authors mention in the Discussion section, information about prescription of MS drugs is lacking. Different drugs have different side effect and complication rates, including severe reactions that could require hospital admissions. Could possible systematically differences in the prescription of MS drugs between the ethnic groups explain some of the skewness between outpatient visits and hospitalizations?

Response: Thank you for this interesting comment. We do not have the data to exclude whether there could have been a systematic difference in prescriptions across the immigrant versus long-term resident population. We have added mention of this as a limitation to the Discussion. However, disease modifying therapies are covered by the government (Exceptional Access Program) in Ontario for those without private drug insurance. Therefore, given coverage and the access patterns to outpatient neurology observed in this study we think that it is unlikely that there would have been a major difference in prescription patterns across groups. Furthermore serious adverse events requiring hospitalization on MS drugs are quite rare.

---

## [Decision Letter · Decision Letter 1]

4 Jun 2020

Health service utilization in immigrants with multiple sclerosis

PONE-D-20-05380R1

Dear Dr. Rotstein,

We’re pleased to inform you that your manuscript has been judged scientifically suitable for publication and will be formally accepted for publication once it meets all outstanding technical requirements.

Kind regards,

Astrid M. Kamperman

Academic Editor

PLOS ONE

Additional Editor Comments (optional):

Reviewers' comments:

Reviewer's Responses to Questions

**Comments to the Author**

1. If the authors have adequately addressed your comments raised in a previous round of review and you feel that this manuscript is now acceptable for publication, you may indicate that here to bypass the “Comments to the Author” section, enter your conflict of interest statement in the “Confidential to Editor” section, and submit your "Accept" recommendation.

Reviewer #1: All comments have been addressed

Reviewer #2: All comments have been addressed

2. Is the manuscript technically sound, and do the data support the conclusions?

Reviewer #1: Yes

Reviewer #2: Yes

3. Has the statistical analysis been performed appropriately and rigorously? 

Reviewer #1: Yes

Reviewer #2: Yes

4. Have the authors made all data underlying the findings in their manuscript fully available?

Reviewer #1: No

Reviewer #2: Yes

5. Is the manuscript presented in an intelligible fashion and written in standard English?

Reviewer #1: Yes

Reviewer #2: Yes

6. Review Comments to the Author

Reviewer #1: (No Response)

Reviewer #2: The authors have answered all questions to my satisfaction and in my view the paper should now be accepted for publication.

7. PLOS authors have the option to publish the peer review history of their article (what does this mean?). If published, this will include your full peer review and any attached files.

Reviewer #1: No

Reviewer #2: No

---

## [Editor Report · Acceptance letter]

25 Jun 2020

PONE-D-20-05380R1 

Health service utilization in immigrants with multiple sclerosis 

Dear Dr. Rotstein:

I'm pleased to inform you that your manuscript has been deemed suitable for publication in PLOS ONE. Congratulations! Your manuscript is now with our production department. 

Kind regards, 

on behalf of

Dr. Astrid M. Kamperman 

Academic Editor

PLOS ONE